# Hard Anodization Film on Carbon Steel Surface by Thermal Spray and Anodization Methods

**DOI:** 10.3390/ma14133580

**Published:** 2021-06-26

**Authors:** Pao-Chang Chiang, Chih-Wei Chen, Fa-Ta Tsai, Chung-Kwei Lin, Chien-Chon Chen

**Affiliations:** 1School of Dentistry, College of Oral Medicine, Taipei Medical University, Taipei 11031, Taiwan; m204095009@tmu.edu.tw; 2Dental Department, Wan Fang Hospital, Taipei Medical University, Taipei 11696, Taiwan; 3Research Center of Digital Oral Science and Technology, College of Oral Medicine, Taipei Medical University, Taipei 11031, Taiwan; 4Division of Neurosurgery, Department of Surgery, Chi Mei Medical Center, Tainan 71004, Taiwan; awei921@gmail.com; 5Department of Occupational Safety and Health, Institute of Industrial Safety and Disaster Prevention, College of Sustainable Environment, Chia Nan University of Pharmacy and Science, Tainan 71710, Taiwan; 6Department of Mechanical Engineering, National United University, Miaoli 36003, Taiwan; fatatsai@nuu.edu.tw; 7Department of Energy Engineering, National United University, Miaoli 36003, Taiwan

**Keywords:** carbon steel, AISI 1045, corrosion, hardness, thermal spray, anodization

## Abstract

In this paper, we used two mass-produced industrial technologies, namely, thermal spraying and anodization methods, to enhance the surface characteristics of AISI 1045 medium carbon steel for use in special environments or products. The anodic film can effectively improve the surface properties of carbon steel. A sequence of treatments of the carbon steel substrate surface that consist of sandblasting, spraying the aluminum film, annealing, hot rolling, cleaning, grinding, and polishing can increase the quality of the anodized film. This paper proposes an anodization process for the surface of carbon steel to increase the corrosion resistance, hardness, color diversification, and electrical resistance. The resulting surface improves the hardness (from 170 HV to 524 HV), surface roughness (from 1.26 to 0.15 μm), coloring (from metal color to various colors), and corrosion resistance (from rusty to corrosion resistant). The electrochemical corrosion studies showed that the AISI 1045 steel surface with a hard anodized film had a lower corrosion current density of 10^−^^5.9^ A/cm^2^ and a higher impedance of 9000 ohm than those of naked AISI 1045 steel (10^−4.2^ A/cm^2^ and 150 ohm) in HCl gas.

## 1. Introduction

Carbon steel is the earliest and most used basic material in modern industry. At present, the output of carbon steel accounts for about 80% of the total steel output in the word. However, the lack of corrosion resistance, poor electrical insulation, low hardness, and colorlessness of carbon steel are unsuitable for some special engineering applications. The surface of carbon steel can be treated with coatings, which protect the steel structure by preventing its contact with a corrosive medium through what is known as barrier protection. A barrier film is widely employed to protect steel in aggressive environments. To improve the characteristics of a steel surface, two popular industrial technologies that can be mass-produced—thermal spraying and anodization technology—can be combined to strengthen the application value of carbon steel surfaces. In contrast to common coating technologies—such as paint, vapor deposition, electro-deposition, or electrophoresis—which can provide only a thin film on the target substrate, thermal spraying can efficiently, provide a thick protective film on a target substrate. In the main thermal spraying process, molten, or semi-molten metal particles are propelled towards the substrate by a stream of air to cause layer-by-layer deposition until the required thickness of the coating is achieved [1,2]. According to Wolczynski’s report [3], the rapid solidification method, such as D-Gun spraying onto the steel, may generate some intermetallic compounds/phases and metastable solidification. Some further modifications of the Jackson and Hunt theory consider the stability of the eutectic growth with the fulfillment of mechanical equilibrium at the triple point of the s/l interface [4].

Anodic oxide film, which is self-assembled and grows from its parent substrate, gives the surface of aluminum alloy high hardness and very good protection properties. There are many potential applications for anodization. For example, oxide film produced on aluminum by anodization is used for corrosion resistance, wear resistance, electrical resistance, and decoration [5,6,7,8,9].

The thermal spraying of metals, oxides, composites, and ceramics has been used on the AISI 1045 medium carbon steel [10] to improve its corrosion resistance, sliding friction, surface hardness, lubrication, high temperature resistance, and anti-sticking applications in biomedical implants [11], the automobile industry [12], electronics [13], and medical tools [14]. The bonding force between Fe and Al is very strong because the intermixing and compound layers form during the thermal treatment [15,16,17]. Thermal spray, due to its rapid solidification nature of the process, deposit evolution is also complicated and commonly leads to ultrafine-grained and metastable microstructures. Wolczynski et al. reported the rapid solidification of the droplets and addressed that the droplets remaining inside the solid shell may be subjected to the slower solidification [18].

Ceramic coatings produced by thermal spraying are widely used in a range of industrial applications to provide wear and erosion resistance, corrosion protection, and thermal insulation to metallic substrates. There is a large demand for the surface treatment of hard anodization on semiconductor equipment because such equipment, such as the components of a vacuum chamber, can react with halogen-containing plasma, leading to corrosion of the components and contaminants from the chamber affecting the wafer substrate. Therefore, corroded components (an electrostatic chuck, a focus ring, an edge ring, a plasma confinement ring, a substrate support, a baffle, a gas distribution plate, a gas distribution ring, a gas nozzle, etc.) must be frequently replaced or removed from the chamber and cleaned, resulting in undesirable chamber downtime [19]. The corrosion resistance of a chamber wall or component can be improved through the formation of a protective film on the component’s surface. For example, a coating can be applied to a chamber wall surface by thermal spraying [20,21,22,23,24]. However, the thermal expansion mismatch between the coating and the chamber or component structure may cause the coating to peel off. Particles may fall off in the chamber, thereby exposing the substrate to corrosion, or loose coating materials may become a problem due to the high melting point of aluminum oxide being directly sprayed on the chamber wall or component.

The hardness and anti-corrosion of steel surface resisting plastic deformation, corrosion, indentation, penetration, and scratching can be achieved by many methods including thin film (for example, PVD, CVD, carburizing, nitriding, electro-deposition, etc.) or thick film (for example, baking paint, electrophoresis, or thermal spray) coatings. In the above method, carbon steel surface can get thick oxide film with high efficiency by thermal spray. However, the porous structure of thermally sprayed oxide film may reduce corrosion resistance on carbon steel surface.

In this research, passive corrosion protection is provided by anodization on the thermally sprayed aluminum film. The anodic layer serves as a barrier to degradation of the metal surface and prevents contact between aggressive electrolytes and the underlying metal substrate. Due to the porous structure of anodized film, a subsequent sealing treatment step is required. The resulting anodic film can have a beautiful, multi-colored surface. In our work, anodic protective film was deposited on a carbon steel surface by plasma thermal spraying and anodization methods. The resulting protective film can effectively improve the color, hardness, electrical insulation, and corrosion resistance of the carbon steel surface. There are some applications for AISI 1045 surface with an anodic film. For instances, high strength roller needs carbon steel as base metal and hard surface. Whereas transport part needs a high-quality carbon steel surface with a dense color oxide film. This paper presents an in-depth study on the effect of current density on the quality of sprayed aluminum after anodization, including the microstructure, film thickness, hardness, surface roughness, and anodization efficiency.

## 2. Materials and Methods

The anodic films, including mild and hard anodic films, were formed on commercial hot-rolled medium-carbon steel sheet, designated AISI 1045 (sample sized: 100 mm diameter × 5 mm, China Steel, Kaohsiung, Taiwan) by thermal spraying and anodization methods in the present study. Two circular anodization samples with a size of 16.4 and 50.0 cm^2^ were used for film thickness study and corrosion test, respectively. 

The experimental procedures can be primarily categorized into eleven stages: (a) Deposition of Al (99.5 wt % Al + 0.5 wt % Mg) film on AISI 1045: aluminum film with 540 μm thickness on AISI 1045 steel (size of φ 4 inch and 5 mm thickness) by electric arc (EA) thermal spraying (200 A, 27 V, 150 mm spray distance, 6 kg air pressure, 30 min spraying time); (b) Densification of sprayed film: thickness of sprayed film decreased from 540 μm to 250 μm through hot rolling process at 200 °C; (c) Pre-treatment for anodization: sample annealing (120 °C, 1 h) in air furnace, mechanical grinding to remove any obvious scratches (#240, #600, and #1200 SiC sandpapers), polishing of surface (1 μm Al_2_O_3_ polishing powder). (d) Anodization: mild anodization (10 vol % H_2_SO_4_, 25 °C, 20 V, 20 min) or hard anodization (5 vol % H_2_SO_4_ + 6 vol % C_2_H_6_O_2_ + 2.4 wt % C_2_H_2_O_4_, at −4 °C, at final voltage of 75 V) for formation of a high hardness and anti-corrosion film on the substrate; (e) Acidification: removing the residual electrolyte from the anodized film (H_2_O, 60 °C, 5 min); (f) Colorization: colorization of mild anodized film in dyeing solution at 50 °C for 5 min; (g) Sealing: increasing the anodic film hardness and anti-corrosion property by water sealing (H_2_O, 100 °C, 40 min); (h) Polishing: increasing the anodic film surface property by mechanical polishing (0.3 μm Al_2_O_3_ polishing powder); (i) Measurement and testing of the anodized film properties of roughness, micro-structure, hardness, thickness, and corrosion resistance; (j) Anodization efficiency calculation; (k) Proposal of suitable anodization parameters for thermally spraying Al on AISI 1045. The related experiment procedure is illustrated in Figure 1. At least 12 samples (2” or 4”) were prepared in one batch for either mild or hard anodization. Several batches were performed to obtain the required samples. The surface roughness (R_a_) values of the work-piece after the multiple treatments ranged from 0.15 to 5.25 μm, as listed in Table 1. The roughness value of each sample (mean ± standard deviation) is measured at 5 different locations.

The microstructures of the sample were characterized by optical microscopy (OM, Nikon LV 150, Nikon Co., Tokyo, Japan) and scanning electron microscopy (SEM, 10 keV, HITACHI Regulus 8100, Hitchi Co., Tokyo, Japan) and energy dispersive X-ray spectroscopy (EDS), and element mapping. The hardness data were obtained with a Vickers diamond micro-hardness tester (Vickers innovative automatic tester, Matsuzawa Co., Ltd, Akita Prefecture, Japan) with a loading of 100 g for AISI 1045 and 1000 g for hardened anodic film. Ten measurements at 10 different sites near the joint regions were performed. The electrochemical behavior was tested with AC impedance and Tafel polarization tests (Zahner Impedance Measuring Unit (IM 6), ZAHNER-elektrik GmbH & Co. KG, Kronach, Germany).

## 3. Results and Discussion

Because the high differences in temperature between the spray layer and the AISI 1045 carbon steel substrate can adversely affect the adhesion to the surface of the substrate, a thermal spray layer can easily crack and peel. Improving the adhesion of thermal spray layers generally requires roughening of the substrate surface. Sandblasting (# 80 Al_2_O_3_) was performed on the AISI 1045 to increase the substrate roughness. The thermal spray layer has a lamellar structure with a large amount of pores and a rough surface; however, the electrolyte can penetrate into the pores and reach the carbon steel, causing a short circuit effect during anodization. Direct anodization of porous as-sprayed Al results in a poor anodic film. To densify the thermally sprayed Al film for the subsequent anodization, the sprayed film was hot rolled to decrease the thickness from 250 μm to 155 μm, a 38% reduction, and the surface roughness was decreased from 5.25 μm to 1.84 μm, a 65% reduction. In addition, the thermally sprayed film was embedded on the AISI 1045 after hot rolling. The details are provided in Figure 2.

The hardness and corrosion resistance of carbon steel can be improved in many ways, such as by thermally spraying ceramics or oxides on carbon steel. As compared to thermally sprayed ceramic or oxide layers, thermally sprayed Al that is subsequently anodized has the advantages of a lower temperature thermal spray and coloring of the anodic film. To make a good quality anodic film on AISI 1045 carbon steel, an anodization mold was designed and fabricated, as shown in Figure 3. The present anodization process using an anodization mold has the following three main features: (a) the anodization mold with an anodization reaction area (4 inches in diameter) is made of Teflon and uses copper as a conductor; (b) the exploded view includes an up-cover with an anodization reaction area, a silicone ring as the water sealing material, a copper plate and rod as conductivity conductors, and an aluminum sheet set between the silicone ring and copper plate; and (c) multiple molds can be in the electrochemical bath in the same time.

In general, industrial anodization processes include multiple steps: degreasing, alkaline etching, anodization, electrolysis coloring, electrophoresis, sealing, and air drying. In our study, the mild anodization steps were simplified to (a) put an AISI 1045 steel work-piece into a sulfuric acid bath (25 °C, 20 V, 20 min) and anodized the thermally sprayed Al which on the steel surface; (b) moving the work-piece (steel surface with thermally sprayed Al and anodization film) into a deacidification bath (H_2_O, 60 °C, 5 min); (c) moving the work-piece into a dyeing bath (50 °C for 5 min), moving the work-piece into a sealing bath (H_2_O, 100 °C, 40 min); and (d) polishing the work-piece (0.3 μm Al_2_O_3_ polishing powder). After mild anodization and dyeing process, the AISI 1045 medium carbon steel surface can present a colored anodic film, as shown in Figure 4.

Hard anodization differs from mild anodization in terms of its lower temperature, higher anodic voltage, longer anodic time, and thicker anodic film. The film thickness of hard anodic film is usually in excess of 50 μm. However, the thickness of the film exceeds the normal limit and causes the phenomena of squeezing and a rough surface. Such anodic film often needs to be mechanically processed or polished again, and then the film is polished to remove about 2 μm from the thickness to smooth the film surface. In addition, the hardness of the film is greater than it was before the grinding, and the lubricity is better.

The process steps of hard anodization include: (a) softening (H_2_SO_4_, 50 °C), (b) stripping (NaOH, 75 °C), (c) chemical cleaning (HNO_3_, 25 °C), (d) chemical polishing (H_3_PO_4_, 95 °C), (e) chemical roughening (NH_4_F 25 °C), (f) anodization (H_2_SO_4_, −4 °C), (g) deacidification (H_2_O, 60 °C), (h) sealing (H_2_O, 100 °C), (i) drying (50 °C), etc. The hard film anodization in this study simplifies the above steps. The main steps include: (1) putting the pre-treatment work-piece into a hard anodization bath containing sulfuric acid + oxalic acid + ethylene glycol (6 vol % H_2_SO_4_ + 2.4 wt % C_2_H_2_OH + 5 vol % EG) under hard anodization parameters of 1.34–9.38 A/dm^2^, 75 V, −4 °C, and 40–100 min in an electrochemical mold; (2) moving the anodized work-piece into a deacidification bath (H_2_O, 60 °C, 5 min); (3) moving the deacidified work-piece into a sealing bath (H_2_O, 100 °C, 40 min); and (4) polishing the work-piece surface (0.3 μm Al_2_O_3_ polishing powder). Figure 5 shows actual photos of the surface color change from silver to black on AISI 1045 steel/thermally-sprayed Al during hard anodization at (a) 1 min, (b) 10 min, (c) 20 min, (d) 0.5 h, (c) 1 h, and (f) 1.5 h at −4 °C and a final voltage of 75 V.

The parameters that need to be controlled for hard anodization include the electrolyte composition, electrolyte temperature, applied voltage, current density (i.e., ampere per square decimeter, abbreviated as ASD thereafter), and anodic time. In this study, we focused on applying different current densities (for example, 1.34 ASD, 2.75 ASD, 3.13 ASD, 5.13 ASD, 7.10 ASD, and 9.38 ASD resulting from 0.22 A, 0.45 A, 0.51 A, 0.84 A, 1.16 A, and 1.54 A in a sample size of 16.4 cm^2^) and finished a detailed study of the influence of current density on anodic film thickness and anodization efficiency at −4 °C and a final voltage of 75 V for 40 to 100 min. To prevent partial burning of the anodic film due to excessive instantaneous current, the current density value to the set value was increased within 20 min and maintained at a specific current density until the final voltage of 75 V. The experimental data such as anodic time, voltage, and current density values were recorded when the anodization was completed. Figure 6 shows the current and voltage curves of hard anodization in a sample size of 16.4 cm^2^. The figure shows that the current value is extremely low when the voltage value is lower than 22 V of the 1.34 ASD curve and 40 V of the 9.38 ASD curve, and the current value can be increased when the voltage value rises to 22 V and 40 V, which shows that the voltage of the hard anodizing treatment must be higher than 22 V with a lower current density or 40 V with a higher current density.

Figure 7 shows the voltage and time (V-T) curves of hard anodization with current densities of 1.34, 2.75, 5.13, 7.10, and 9.38 ASD, respectively. The anodization time was based on a final voltage of 75 V that is commonly used in practical application. Based on the Figure 7, a lower current density applied to thermally sprayed Al will require a longer anodization time to reach the final voltage of 75 V. For instances, a current density of 1.34 ASD takes 108 min to the final voltage. On the other hand, a higher current density applied to thermally sprayed Al spend a shorter anodization time to the final voltage of 75 V. For instances, a current density of 7.10 ASD only needs ~32 min (the shortest among the examined current densities) and the others require ~46–50 min to the final voltage of 75 V.

According to Faraday’s law, the relationship between anodic current and film thickness can be expressed as Equation (1), which can be further simplified to Equation (2):Q = It = nFN = nFρDA/M (1)
D = ItM/nFρA(2)
where Q is charge (C), I is current (A), t is time (sec), n is valence (3), F is Faraday constant (96,500), N is mole, ρ is density (2.4 g/cm^3^), D is thickness (cm), A is surface area (cm^2^), and M is molecular weight (78 g/mol).

In this study, the anodization efficiency sample area was 16.4 cm^2^ for the test of film formation. Therefore, Equation (2) can be simplified to Equation (3), and the anodization efficiency can be calculated using Equation (4), where D_a_ and D are experimental thickness and calculation thickness, respectively. The efficiency (η in %) related to charge equation can be determined by Equation (5).
D (μm) = 0.0702Q (3)
η (%) = 100 × D_a_/D(4)
η (%) = 100 × D_a_/0.0702Q(5)

Based on the Equation (5), D_a_ is depended on the experimental current density (ASD), the efficiency related to charge can be plotted as Figure 8. It can be noted in Figure 8, generally, the smaller anodization current density the higher anodization efficiency.

In practical anodization, current is applied and current density is calculated due to different sample size. Also in practical anodization, time (indicating the cost) is an important factor. Figure 9 shows the current and time (I-T) curves of hard anodization with current densities of 1.34, 2.75, 5.13, 7.10, and 9.38 ASD to reveal the efficiency of anodization. Based on the final voltage of 75 V, the anodization time and the film thickness were different for various ASDs. Based on Equations (1)–(4) and measurement of the film thickness, the anodization efficiency under different current densities can be calculated. The figure and calculation results showed that when the current density is smaller, such as 1.34 ASD, the anodization efficiency can reach 81.4%, but the anodization treatment time is longer, such as 100 min. On the other hand, when the current density is higher, such as 9.38 ASD, the anodization efficiency decreases to 36.5%. The experiment results also showed that a suitable current density is between 2.75 to 7.10 ASD, and the efficiency of anodizing is about 60%. During the anodic treatment, the anodic film grows between the anodization film and the aluminum interface, but the film also dissolves between the anodic film and the electrolyte interface. Excessive current density will easily cause a local temperature spike on anodic film, which will increase the dissolution rate of the anodic film and reduce the anodization efficiency.

Figure 10 shows the hard anodic film on thermally sprayed Al/AISI 1045 carbon steel. This photo also shows that the sprayed aluminum film has been rolled, and it can indeed prevent the electrolyte from penetrating through the surface to the carbon steel substrate. Figure 10a shows that the hard anodized film still has many holes and defects, and Figure 10b shows that the microstructure is more obvious on AISI 1045 carbon steel after 5 vol % nitric acid + alcohol solution etching and on Al/anodic film after 5 wt % NaOH solution etching. It can be seen that the sprayed aluminum and the hard anodized film have large numbers of defects, although the hard anodized film has a larger number. However, the high hardness and corrosion resistance of the anodic film are sufficient to provide protection for the surface of carbon steel. Figure 10c presents a detailed photomicrograph of the thermally sprayed aluminum on the surface of AISI 1045 carbon steel substrate after hard anodizing. The thickness of the sprayed aluminum after rolling was about 250 μm. After the hard anodizing, the sprayed aluminum consumed 75 μm of the thickness and grew into a 175 μm hard anodized film. The thickness of the total film increased by about 100 μm, an increase of 133%. This was the result of the volume expansion caused by the cell structure of the anodized film.

Figure 11 shows microscopic images of the interface of the carbon steel substrate and sprayed aluminum, which may be used to evaluate the ability of the sprayed aluminum to block the penetration of electrolytes onto the surface of the steel under different current densities. It should be pointed out that there may still have defects in the thermally sprayed and hot-rolled Al film. If the applied current density is too high, it may induce poor anodization film with low hardness. The applied current density values during anodizing were (a) 1.34 ASD (A/dm^2^), (b) 2.75 ASD, (c) 3.13 ASD, (d) 5.13 ASD, (e) 7.10 ASD, and (f) 9.38 ASD. As compare to those of uncoated AISI 1045 steel (170 ± 8 HV**_0.1_**), the hardness values were 524 ± 23, 443 ± 17, 421 ± 20, 408 ± 20, 353 ± 14, and 302 ± 16 HV_1.0_, under current densities of 1.34, 2.75, 3.13, 5.13, 7.10, and 9.38 ASD, respectively. This phenomenon also means that too high a current density will make the electrolytes penetrate into the sprayed film and reach the carbon steel surface, causing the anodization to fail.

Figure 12 shows an SEM image of the AISI 1045 carbon steel substrate surface after thermal spraying of aluminum and hard anodization. Figure 12a shows the AISI 1045 carbon steel/thermal sprayed aluminum/hard anodized film stack structure. Figure 12b presents SEM mapping of the AISI 1045 steel. The representative element of the steel is Fe, that of the thermal spray is Al, and that of the hard anodized film is O. In the image, the Al-Fe interface has a small amount of O, which can be ascribed to local oxidation of the surface of the carbon steel substrate during the arc spraying process. Figure 12c–e presents individual oxygen (O), aluminum (Al), and iron (Fe) SEM mapping of Figure 12b, respectively.

Carbon steels are used as the preferred construction materials across industries and are considered more economical than their costly corrosion-resistant alloy counterparts. It is necessary to enhance the corrosion resistance of carbon steel for actual applications. Electrochemical anodization is a new surface treatment technique for aluminum film on carbon steel. The corrosion resistance of aluminum can be greatly increased by forming an oxide layer through anodization. Therefore, an anodic film on carbon steel can efficiently enhance the corrosion resistance of carbon steel. 

It should be pointed out that the AISI 1045 steel with thermally sprayed Al and anodization with various ASD exhibited excellent anti-corrosion performance. No corrosion phenomenon can be observed after testing by typical 3.5 wt % NaCl solution. In order to reveal the anti-corrosive ability, the test samples and saturated hydrochloric acid (HCl) solution were placed in a closed container for various duration. Figure 13 presents actual photos comparing AISI 1045 carbon steel with (left side) and without (right side) hard anodized film in HCl gas atmosphere after (a) 30 sec, (b) 0.5 h, (c) 10 h, and (d) 24 h. The steel surface with the anodized film protection remained the original black, hard anodized film. In contrast, the naked steel surface began to rust, and the surface color changed to brown as the corrosion time increased.

Figure 14 presents the results of the electrochemical behavior test with (a) Tafel polarization and (b) AC impedance on a naked AISI 1045 carbon steel surface and the surface with a hard anodized film in HCl gas atmosphere. In Figure 14a, the naked steel surface has a higher corrosion current density (10^−4.2^ A/cm^2^) than the 10^−^^5.9^ A/cm^2^ of the steel surface with hard anodic film protection. For the AC impedance characteristics, Figure 14b presents a Bode plot showing that, after anodization, the impedance of the AISI 1045 carbon steel (9000 ohm) was higher than that of the naked steel (150 ohm) in the lower frequency range. Figure 14c,d Nyquist plots also show that the impedance of the AISI 1045 carbon steel was higher after anodization. Figure 14e showed the schematic diagram of EIS testing sample of carbon steel/spray Al/anodization film/solution structure. Figure 14f shows the typical model equivalent circuit of a rust or oxide film on the metal [25], the subscript of rust means rusts on AISI 1045, and the subscript of anodization means anodic film on AISI 1045. R_so_, C_dl_, R_ct_, C_rust_, C_anodization_, R_rust_, and R_anodization_ were denoted to solution resistance, double layer capacitance, charge transfer resistance, rust capacitance, anodic film capacitance, rust resistance, and anodic film resistance, respectively. Based on the AC impedance rule and simulation [25], R_so_ is 4 ohm, C_dl, rust_ is 83 pF, R_ct, rust_ is 7 ohm, whereas C_dl, anodiction_ is 235 pF, and R_ct, anodization_ is 12 ohm. All of above values are very small and imply that the R_so_, C_dl_, and R_ct_ have limited influence on impedance. From the Nyquist plot of Figure 14c, we can derive R_rust_ and C_rust_ is 135 ohm and 0.67 μF, respectively. Whereas, as revealed by the Nyquist plot of Figure 14d after anodization, R_anodization_ and C_anodization_ reaches 12,000 ohm and 15 μF, respectively. According to the results shown in Figure 14c,d, the AISI 1045 surface resistance increased from 135 ohm (rust) to 12,000 ohm (anodization) after anodization and the main contribution of surface impedance were R_rust_ and R_anodization_. This indicates that the surface corrosion resistance of AISI 1045 can be significantly improved after anodization. Figure 14e showed structure of thermally sprayed Al film and anodization on AISI 1045 corresponding to resistance and capacitance elements, respectively. Figure 14f revealed that the structure can be simulated by Equivalent circuit.

## 4. Conclusions

The AISI 1045 medium carbon steel surface was coated with an aluminum film by thermal spraying, and then the aluminum film was converted to an oxide film by anodization. Hot rolling densifies the thermally sprayed Al film on the steel surface and improves the effects of anodization. Following mild anodization and dyeing enable Al-coated steel surfaces with various colors. Whereas hard anodization delivers carbon steel surfaces with high hardness and excellent corrosion resistance. Compared to Al-coated carbon steel surface with hard anodic film, the naked steel surface began to rust and color changed to brown after 30 min in HCl gas. Whereas all the Al-coated steel with anodization films can indeed improve the corrosion resistance of steel surface. No corrosion phenomenon can be observed after 24 h in HCl gas environment. With an applied current density of 1.34 ASD, the Al-coated steel carbon steel surface exhibited a higher hardness of 524 HV_1.0_, a lower corrosion current density of 10^−5.9^ A/cm^2^, and a higher surface resistance of 12,000 ohm compared to those of naked AISI 1045 steel with a 170 HV_0.1_, 10^−4.2^ A/cm^2^ and 135 ohm, respectively. With a suitable anodization film, AISI 1045 carbon steel can exhibit improved surface characteristics such as high hardness, corrosion resistance, and coloring.

## Figures and Tables

**Figure 1 materials-14-03580-f001:**
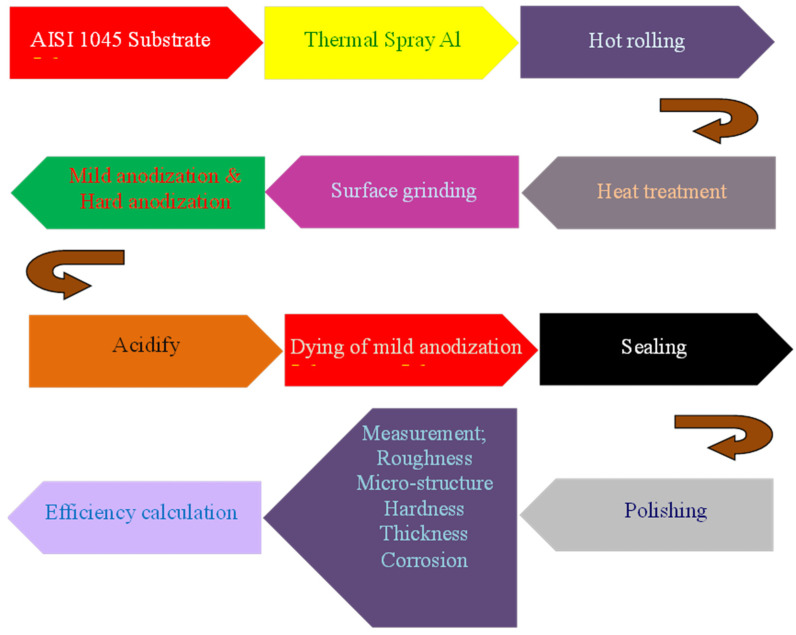
Flow chart illustrating the entire experiment process.

**Figure 2 materials-14-03580-f002:**
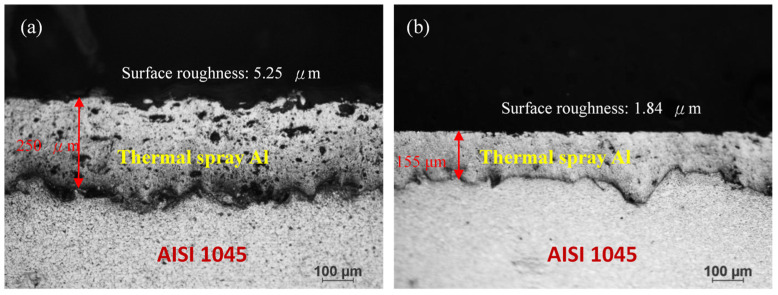
Optical micro-images of thermal spray Al on AISI 1045; (**a**) before hot rolling, and (**b**) after hot rolling.

**Figure 3 materials-14-03580-f003:**
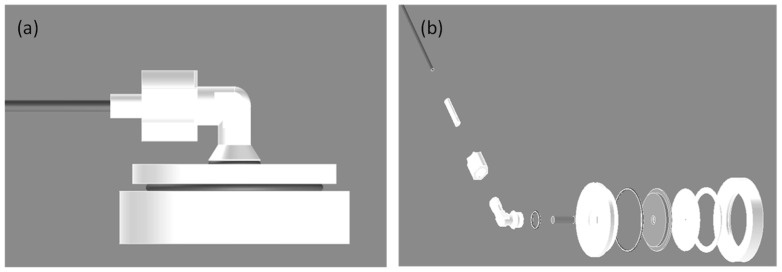
Electrochemical mold for anodization and electrochemical test; (**a**) the combination diagram, (**b**) the exploded diagram.

**Figure 4 materials-14-03580-f004:**
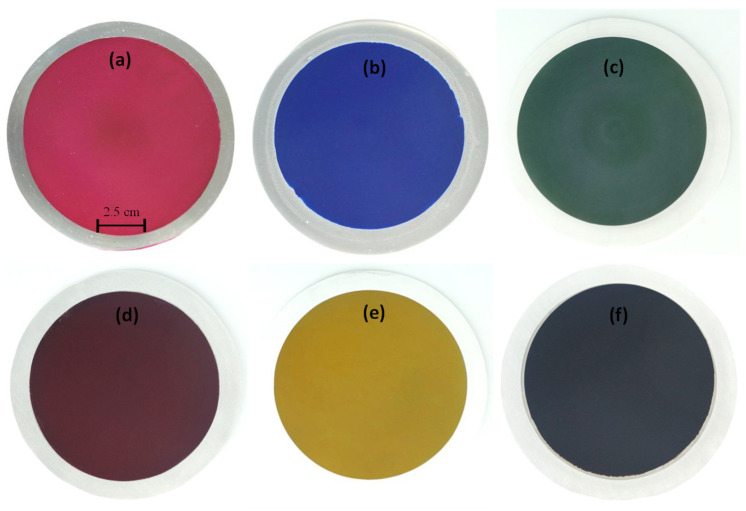
Optical images of thermal spray Al on AISI 1045 after mild anodization and dyed with color: (**a**) red, and (**b**) blue, (**c**) green, (**d**) brown, (**e**) yellow, and (**f**) black.

**Figure 5 materials-14-03580-f005:**
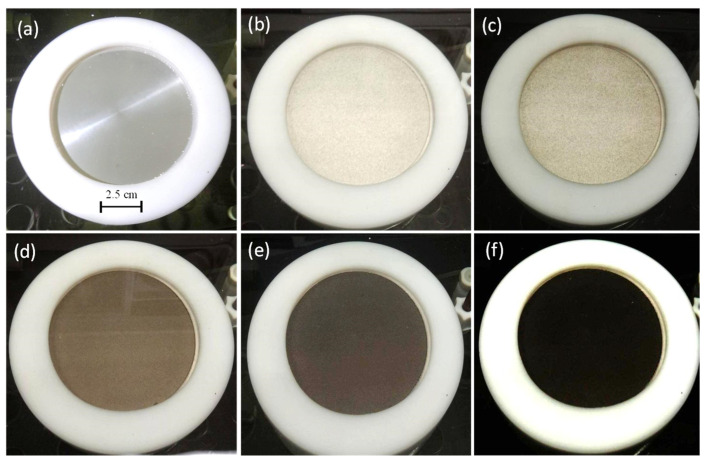
Actual photos of surface color change on AISI 1045 steel/thermal sprayed Al during hard anodization in (**a**) 1 min, (**b**) 10 min, (**c**) 20 min, (**d**) 0.5 h, (**e**) 1 h, and (**f**) 1.5 h.

**Figure 6 materials-14-03580-f006:**
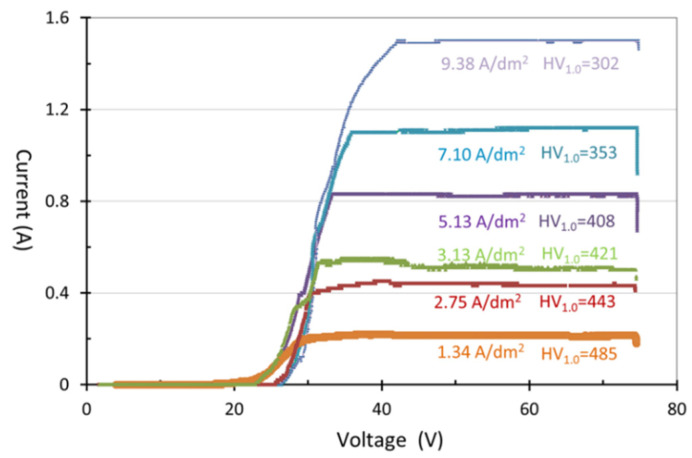
I-V curves of hard anodization with current densities of 1.34, 2.75, 5.13, 7.10, and 9.38 ASD.

**Figure 7 materials-14-03580-f007:**
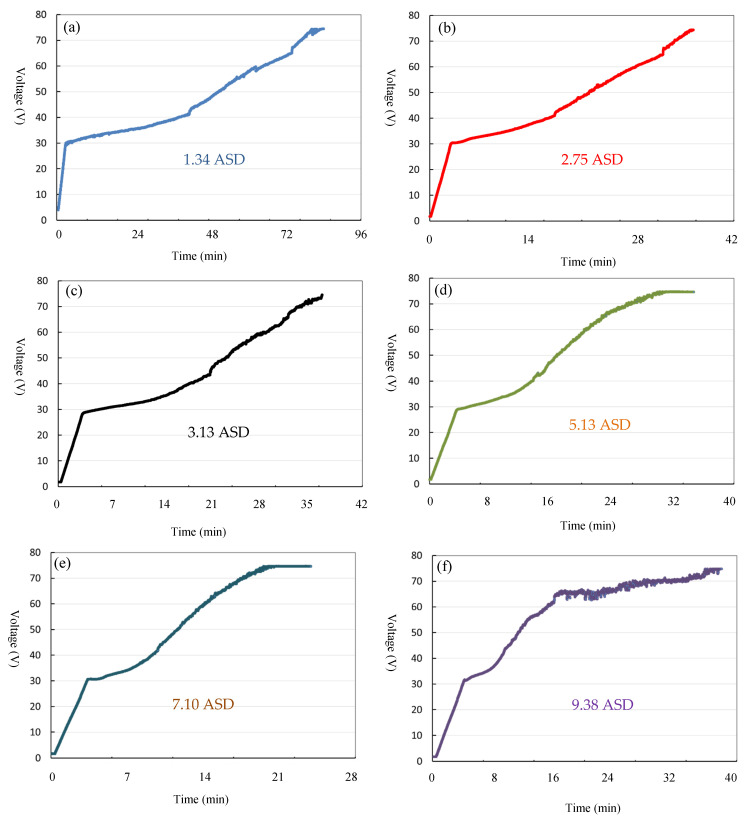
V-T curves of hard anodization with current densities of (**a**) 1.34, (**b**) 2.75, (**c**) 3.13, (**d**) 5.13, (**e**) 7.10, and (**f**) 9.38 ASD, respectively.

**Figure 8 materials-14-03580-f008:**
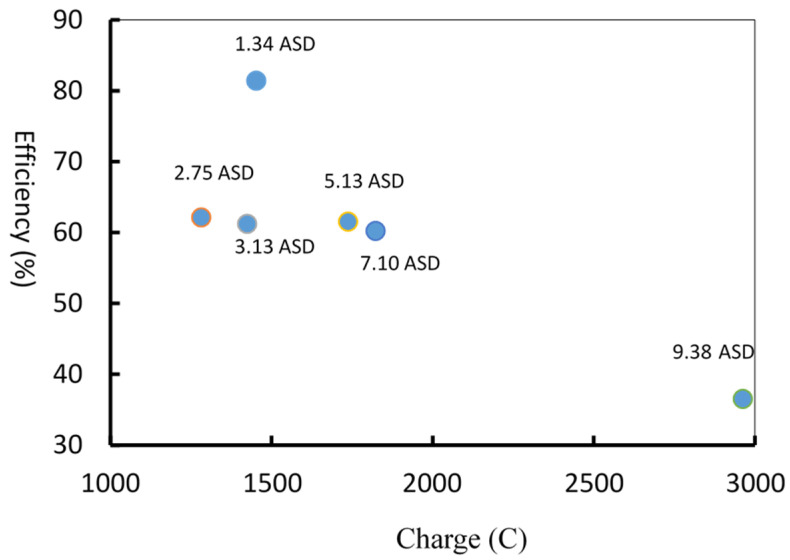
Plot of anodization efficiency related to charge with various current density.

**Figure 9 materials-14-03580-f009:**
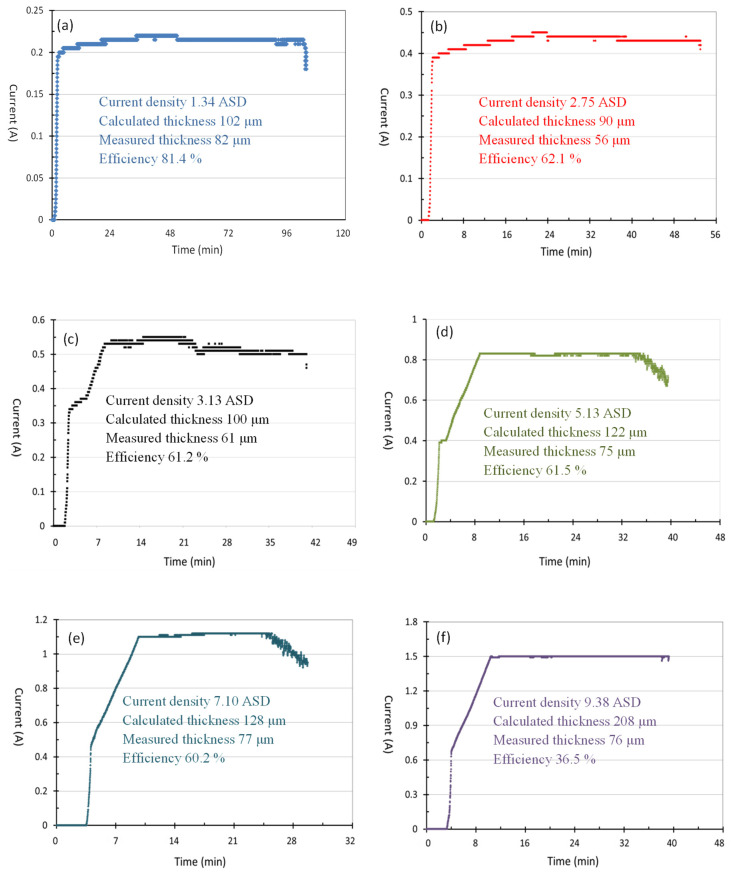
I-T curves of hard anodization with current densities of (**a**) 1.34, (**b**) 2.75, (**c**) 3.13, (**d**) 5.13, (**e**) 7.10, and (**f**) 9.38 ASD.

**Figure 10 materials-14-03580-f010:**
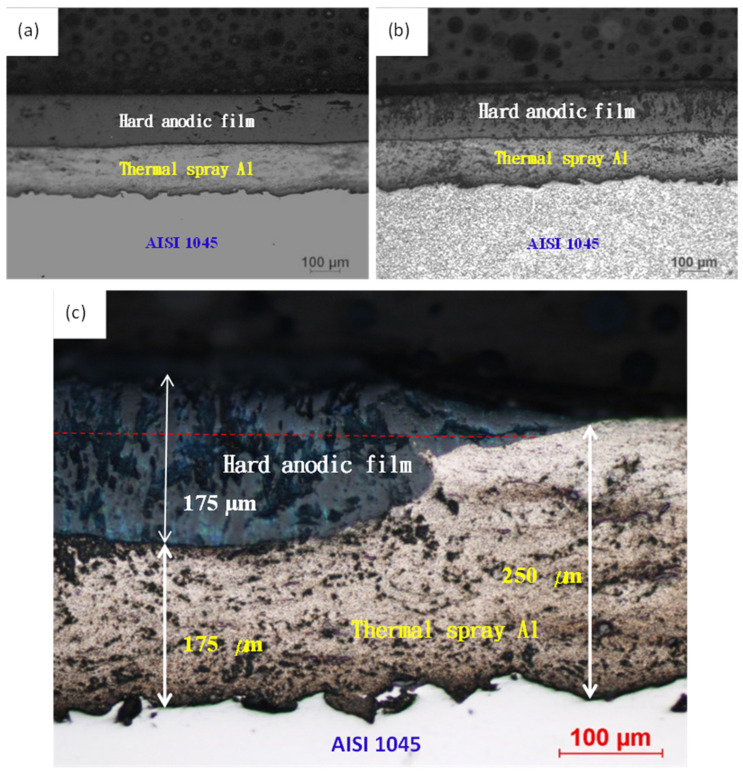
Optical micro-images of thermally sprayed Al on AISI 1045 after hard anodization; (**a**) before etching, (**b**) after etching, and (**c**) partial enlarged view.

**Figure 11 materials-14-03580-f011:**
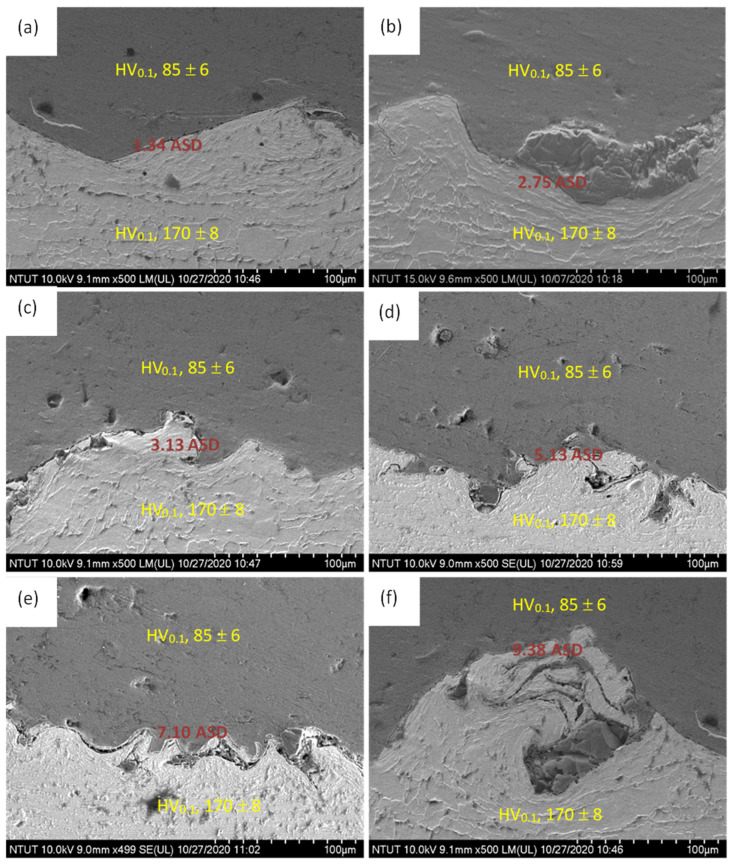
SEM images of AISI 1045/thermally sprayed Al interface after application of current densities: (**a**) 1.34 ASD, (**b**) 2.75 ASD, (**c**) 3.13 ASD, (**d**) 5.13 ASD, (**e**) 7.10 ASD, and (**f**) 9.38 ASD. Microhardness values (HV_0.1_ for AISI 1045 and HV_1.0_ for hardened surface) were given for comparison.

**Figure 12 materials-14-03580-f012:**
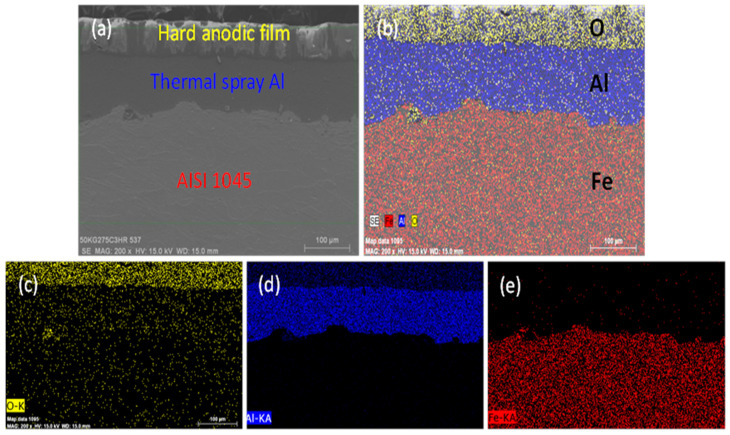
SEM Images of AISI 1045/thermally sprayed Al/hard anodic film interfaces; (**a**) hard anodic film on thermally sprayed Al/AISI 1045, (**b**) SEM mapping of AISI 1045/thermally sprayed Al/hard anodic film with (**c**) O mapping, (**d**) Al mapping, and (**e**) Fe mapping, respectively.

**Figure 13 materials-14-03580-f013:**
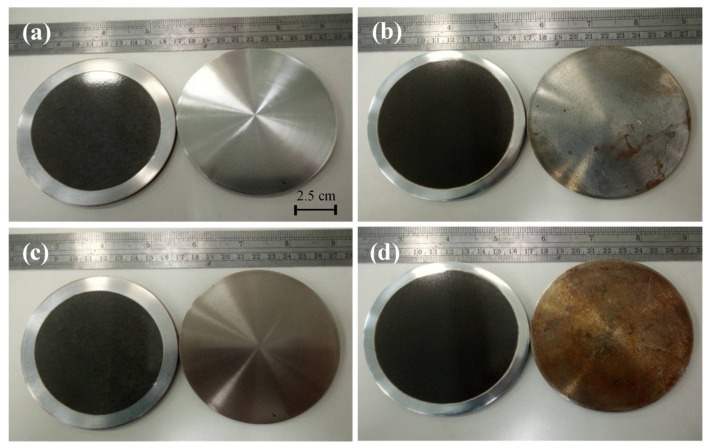
Actual photos of surface changes on AISI 1045 steel with (left side) and without (right side) hard anodization in HCl gas atmosphere after (**a**) 30 sec, (**b**) 0.5 h, (**c**) 10 h, and (**d**) 24 h.

**Figure 14 materials-14-03580-f014:**
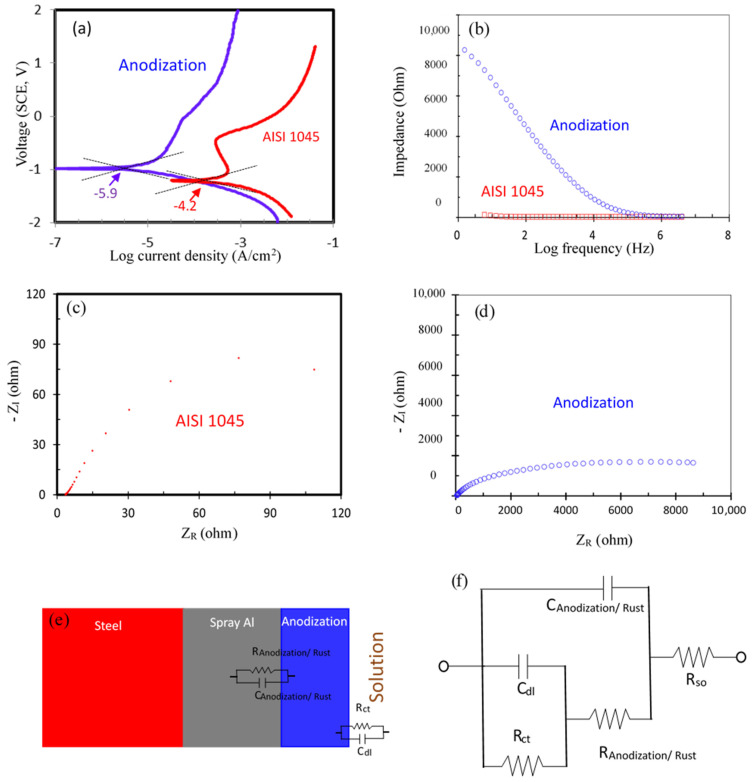
Electrochemical test curves of AISI 1045 and anodization: (**a**) Tafel curve, (**b**) Bode plot, (**c**) Nyquist plot of AISI 1045, and (**d**) Nyquist plot of hard anodized film on AISI 1045, (**e**) structure of films on AISI 1045, (**f**) Equivalent circuit.

**Table 1 materials-14-03580-t001:** Surface roughness of AISI 1045 steel after various surface treatments.

Surface Treatment	Roughness (R_a_, μm)
AISI 1045 Substrate	1.26 ± 0.015
Thermal spray Al	5.25 ± 0.036
Hot rolling	1.84 ± 0.016
Grinding (# 1500)	0.55 ± 0.007
Mild anodization	0.62 ± 0.010
Hard anodization	0.74 ± 0.011
Polished	0.15 ± 0.007

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
