# Peer review of "Hard Anodization Film on Carbon Steel Surface by Thermal Spray and Anodization Methods"

_materials, 2021, doi:10.3390/ma14133580_

Round 1
Reviewer 1 Report
see attached file containing some comments to your manuscript

Author Response
Response to Reviewers
To whom it may concern:
We have attached the revised manuscript titled “Enhance Carbon Steel Surface Characteristics by Thermal Spray and Anodization Methods” by Chen et al. The manuscript is code as materials-1237998.
To improve our manuscript, we have made necessary changes in the revised draft to address every question raised by the reviewers. Accordingly, relevant paragraphs and references in the revised manuscript have been thoroughly updated. Reviewer’s comments and corresponding replies are described below. All the revisions are characters in blue color.
In closing, if there are any corrections that we missed or need to be readjusted, please let us know.
Thank you for your effort to improve the quality of this manuscript.
Sincerely Yours,
Questions and Replies
Reviewer #1:
(1) According to the authors discussion (in the introduction of the delivered manuscript) the molten or semi-molten particles are usually deposited onto the steel substrate. Therefore, I suggest to take into account and comment by the authors the theoretical model for the deposition of the partially melted particles published in the following paper: [ref a] Development of the Jackson and Hunt Theory for Rapid Eutectic Growth, Archives of Metallurgy and Materials, 63(1) (2018) 65-72. DOI: 10.24425/118910. The deposited semi-melted particles present not only the columnar structure (f1+f2 area in the Fig. 1a, [ref a]) but the eutectic structure (lamellar or rod-like eutectic structure) as well. The mathematical description of the non-homogeneous micro-field for the solute concentration, as formed ahead of the eutectic solid/liquid interface, is also delivered in the [ref a] - paper. Moreover, the formation of some phases in the conditions which ensure the metastable solidification is also described in [ref a]. It should be emphasized that in the case of experiment shown in [ref a]. The Fe-40Al particles were settled onto the substrate and in the authors’ experiment the Al – particles, only. Nevertheless, an analogy between both compared experiments exists. The solidification of the Fe-Al particles leads to formation of some micro-joints, as mentioned in the [ref a] – paper. It results in formation of some intermetallic phases or compounds (FeAl - phase, Fe2Al5 – compound). It would be interesting to know whether some intermetallic phases (or compounds) could also be expected at the boundary steel / film in the authors’ experiment, may be in the future investigations. The mentioned phases (or compounds) are high temperature resistant (presents high temperature melting point) while Al-film is not so resistant (lower melting point).Therefore, the steel substrate coated by the Fe-Al particles could be applied to the situation when the material is subjected to high temperature activity. The behavior of the Fe-Al particles (droplets) during their deposition (also motion) onto the substrate is well explained in the additional experiment as described in the [ref a]– paper.
Reply: We thank reviewer’s suggestion for a very good reference. We have read this reference and placed it in our revised manuscript. The sentences are in the introduction session (lines 42-45).
“According to Wolczynski’s report [3], the rapid solidification method, such as D-Gun spraying onto the steel, may generate some intermetallic compounds/phases and metastable solidification. Some further modifications of the Jackson and Hunt theory consider the stability of the eutectic growth with the fulfillment of mechanical equilibrium at the triple point of the s/l interface [4]”.
(2) [ref b] Solidification Mechanism of the D-gun Sprayed Fe-Al Particles, Archives of Metallurgy and Materials, 62(4) (2017) 2391-2397. DOI: 10.1515/amm-2017-0352. In this case, particles (droplets) are deposited onto the water surface, [ref b]. Therefore, these droplets are not damaged (not flattened). Thus, observation and study of the mechanism of solidification were possible. The authors, should refer their experiment (their investigations) to the results contained within the both mentioned papers, [ref a], and [ref b]. Additionally, the authors should also conclude whether the reaction between Fe (steel) and aluminum could be expected (or not) in the case of the Al - film formation during their experiment. This phenomenon plays an essential role since the formation of some intermetallic phases or compounds at the Fe (steel) and Al(film) boundary creates a diffusional connecting between both materials. Usually, the mentioned connection improves the required / desired adhesion.
Reply: Thanks again for the good references. We have revised the relative sentences (lines 53-57).
“The bonding force between Fe and Al is very strong because the intermixing and compound layers form during the thermal treatment [15-17]. Thermal spray, due to its rapid solidification nature of the process, deposit evolution is also complicated and commonly leads to ultrafine-grained and metastable microstructures. Wolczynski et al. reported the rapid solidification of the droplets and addressed that the droplets remaining inside the solid shell may be subjected to the slower solidification [18].”
We hope the revision in the revised manuscript can meet the reviewer’s comments.
Chien Chon Chen, Ph. D.
Professor
Department of Energy Engineering, National United University, Miaoli 36003, Taiwan
ccchen@nuu.edu.tw

Reviewer 2 Report
- The title and abstract do not correlate with the manuscript content.
- The Introduction includes a number of irrelevant references. You should limit to only thermal spraying and aluminium anodization.
- The Introduction does not give any potential application areas where this kind of very complicated processing could be used.
- The Materials and Methods is borderline acceptable, but I doubt if these tests could be replicated with the given information.
- A major reason to reject the current version of the manuscript is that the authors do not provide information of replicate tests or error estimation methods.
- The Results contains a number of illogical statements, like line 140, how can you improve steel hatdness by spraying a coating; line 142, thermally sprayed aluminium has used for corrosion prevention for decades; line 162, you do not have steel surface but aluminium surface; line 192, what is ASD; line 212 Q is not power but charge; line 260, hardness of steel or aluminium; line 291, not steel surface
- Why the compliccated anodisation procedure, did you try to test that the sprayed and rolled surface is suitable for normal and hard anodization?
- Figure 6 is hard to understand, why do you have current as dependent factor and current density as parameter? If x-axis voltage is cell voltage then the figure does not give correct information about sample surface voltage and current.
- Part of equation 1 lacks time as factor
- Fig 7, why current as dependent factor and current density as parameter?
- Fig 12, the Al icorr is about 10-5 not 10-7, Fig 12b should include also the phase angle and with clear scales. For EIS analysis you must provide the equivalent circuit component values. The minium frequency is too high to provide meaningful results.
- Conclusions are repeating the presented results, some of the conclusions are not supported by the provided results.
Author Response
Response to Reviewers
To whom it may concern:
We have attached the revised manuscript titled “Enhance Carbon Steel Surface Characteristics by Thermal Spray and Anodization Methods” by Chen et al. The manuscript is code as materials-1237998.
To improve our manuscript, we have made necessary changes in the revised draft to address every question raised by the reviewers. Accordingly, relevant paragraphs and references in the revised manuscript have been thoroughly updated. Reviewer’s comments and corresponding replies are described below. All the revisions are characters in blue color.
In closing, if there are any corrections that we missed or need to be readjusted, please let us know.
Thank you for your effort to improve the quality of this manuscript.
Sincerely Yours,
Chien Chon Chen, Ph. D.
Professor
Department of Energy Engineering, National United University, Miaoli 36003, Taiwan
ccchen@nuu.edu.tw
Questions and Replies
Reviewer #2:
(1) The title and abstract do not correlate with the manuscript content.
Reply: Thanks for reviewer’s suggestion. We have rewritten the title and abstract.
(2) The Introduction includes a number of irrelevant references. You should limit to only thermal spraying and aluminium anodization.
Reply: Thanks for reviewer’s suggestion. We have removed some of irrelevant references. The Introduction section has been revised.
(3) The Introduction does not give any potential application areas where this kind of very complicated processing could be used.
Reply: Thanks for the suggestion. We have added several sentences about the applications in the Introduction session (lines 76-79).
“There are some applications for AISI 1045 surface with an anodic film. For instances, high strength roller needs carbon steel as base metal and hard surface. Carbon steel needs insulation in electricity usages. Whereas transport part needs a high quality carbon steel surface with a dense color oxide film.”.
(4) The Materials and Methods is borderline acceptable, but I doubt if these tests could be replicated with the given information.
Reply: In this paper, we used two mass-produced industrial technologies, namely, thermal spraying and anodization methods, to modify the surface of AISI 1045. Though the procedures are complicate but it is reproducible. In the Materials and Methods section (lines 101-102), we have added the following sentences.
“At least 12 samples (2” or 4”) were prepared in one batch for either mild or hard anodization. Several batches were performed to obtain the required samples.”
(5) A major reason to reject the current version of the manuscript is that the authors do not provide information of replicate tests or error estimation methods.
Reply: Thanks for reviewer’s suggestion. We have revised the Materials and Methods section (in blue fonts) and hope to meet the requirement.
(6) The Results contains a number of illogical statements, like line 140, how can you improve steel hatdness by spraying a coating; line 142, thermally sprayed aluminium has used for corrosion prevention for decades; line 162, you do not have steel surface but aluminium surface; line 192, what is ASD; line 212 Q is not power but charge; line 260, hardness of steel or aluminium; line 291, not steel surface
Reply: Thanks for reviewer’s information; we have removed the useless statements.
(7) Why the complicated anodization procedure, did you try to test that the sprayed and rolled surface is suitable for normal and hard anodization?
Reply: In order to obtain a high quality anodic film on AISI 1045 the pre-treatment before anodization and post treatment after anodization are needed. We have added one sentence at lines 122-123 to address this issue.
“Direct anodization of porous as-sprayed Al results in a poor anodic film.”
(8) Figure 6 is hard to understand, why do you have current as dependent factor and current density as parameter? If x-axis voltage is cell voltage then the figure does not give correct information about sample surface voltage and current.
Reply: If practical use, a specific current is applied with increasing voltage. The anodization sample size is 16.4 cm2 and will give a desired current density. The explanation is given at lines 177-178.
“(i.e., 1.34 ASD, 2.75 ASD, 3.13 ASD, 5.13 ASD, 7.10 ASD, and 9.38 ASD resulting from 0.22 A, 0.45 A, 0.51 A, 0.84 A, 1.16 A, and 1.54 A in a sample size of 16.4 cm2)”
(9) Part of equation 1 lacks time as factor
Reply: Thanks for the comment. Eqn. 1 should be Q = It = nFN = nFρDA/M, where t is time. Total charge Q (=It) equals to the right two terms that works for anodization on the sample and do not have time factor.
(10) Fig 7, why current as dependent factor and current density as parameter?
Reply: In practical anodization, current is applied and current density is calculated due to different sample size. Also in practical anodization, time (indicating the cost) is important. Thus in Fig. 7, the plots of current vs. time were shown to reveal the efficiency of anodization. We have revised corresponding paragraph in the revised manuscript. (please see lines 202-204)
(11) Fig 12, the Al icorr is about 10-5not 10-7, Fig 12b should include also the phase angle and with clear scales. For EIS analysis you must provide the equivalent circuit component values. The minimum frequency is too high to provide meaningful results.
Reply: Thanks for reviewer’s suggestion. We have added equivalent circuit diagram in Fig. 12(e) and discussed it in the session (lines 281-291). In addition, in Fig. 12(a), the Icorr are 10-4.4 A/cm2 (AISI 1045) and 10-6.5 A/cm2 (anodic film). Corresponding values are marked in Fig. 12(a).
(12) Conclusions are repeating the presented results; some of the conclusions are not supported by the provided results.
Reply: Thanks for reviewer’s suggestion; we have revised the conclusion session in lines 297-304.
We hope the revision in the revised manuscript can meet the reviewer’s comments.

Round 2
Reviewer 2 Report
- The title and paper content do still not match, the paper deals with aluminium coating, not with steel properties.
- The new references do really not deal with the content of the paper.
- The paper still lacks applications, now there are only some generic ones of which the electricity is confusing.
- Materials and Methods section has been improved and is good enough.
- Still some confusing statements: Line 146, you do not anodize steel; Line 149, the surface is not steel surface; Line 210, what is anode film?; Line 218, no hard anodized film on steel but on aluminium; Line 239, not naked but uncoated; Line 253, you do not anodize steel but you coat it with Al that is anodized; Line 297, not densities but densifies, Line 298, not a color of steel but an Al-coated steel; Line 302, one last time, you have not anodized steel.
- State that ASD means ampere per square decimeter.
- Figur 6 is misleading, you should plot current or current density and potential as a function of time.
- Eq 3 and 4, is efficiency related to charge, a plot would be interesting.
- Fig 7 should have identical axes for all plots. What was the reason to use tests with different lengths? Did you aim for about 100 um calculated thickness?
- Lines 238-240 include some odd statements. Do you mean that the sprayed and rolled aluminium layer is about the same before anodization, but anodization itself cause corrosion on the steel/Al interface? In this case the rolled Al must be quite porous. If steel is anodically dissolved you should detect iron on the surface of Al, try the ferroxyl test.
- The caption of Fig 10 is misleading, especially figures c to e.
- Lines 274-275, how can you make electrochemical measurements in gas atmosphere, you need your sample and other electrodes immersed in solution?
- Fig 12 a analysis is done wrong, in Tafle analysis you should extend the linear ranges to corrosion potential and read the intersect. The corrosion current density is not the minimum value of the plot. Fig 12b the y-axis is not showing log(Z) but Z. Fig 12 d the axes must have identical tick lengths so that the plot is not distorted. Fig 12 e is mislabelled, the outer time constant is the Al and the inner one steel.
- EIS analysis: If your component values are in ohms to 100 ohms range, how can you get several kilo-ohms as in Fig 12b? This is possible only if your Rct goes to kilo-ohms. Usually, what you have marked as Rct is actually the resistance in the pores of a coating.
- Your conclusions about corrosion resistance are rather weak, you can expect aluminium to stand better corrosion than steel in HCl environment. The important part that is missing is the effect of different anodization processes on corrosion resistance.
Author Response
Dear Editor and Reviewer:
We have attached the revised manuscript titled “Hard Anodization Film on Carbon Steel Surface by Thermal Spray and Anodization Methods” by Chen et al. The manuscript is code as materials-1237998.
To improve our manuscript, we have made necessary changes in the revised draft to address every question raised by the reviewers. Accordingly, relevant paragraphs and references in the revised manuscript have been thoroughly updated. Reviewer’s comments and corresponding replies are described below. All the revisions are characters in blue color.
In closing, if there are any corrections that we missed or need to be readjusted, please let us know.
Thank you for your effort to improve the quality of this manuscript.
Sincerely Yours,
Chien Chon Chen, Ph. D.
Professor
Department of Energy Engineering, National United University, Miaoli 36003, Taiwan
ccchen@nuu.edu.tw
Questions and Replies
Reviewer #2:
(1) The title and paper content do still not match, the paper deals with aluminium coating, not with steel properties.
Reply: Thanks for reviewer’s suggestion. We have rewritten the title as “Hard Anodization Film on Carbon Steel Surface by Thermal Spray and Anodization Methods”.
(2) The new references do really not deal with the content of the paper.
Reply: According to reviewer 1 suggestion, refs 3-5, 15-18; references 25 is related to thermal spray, and references 22 is related to EIS.
(3) The paper still lacks applications, now there are only some generic ones of which the electricity is confusing.
Reply: Thanks for reviewer’s suggestion, we have removed the electricity application in carbon steel. We have added a paragraph in lines 71-77 about the application and why thermally sprayed Al film and anodization on carbon steel surface are required.
(4) Materials and Methods section has been improved and is good enough.
Reply: Thanks for reviewer’s comment.
(5) Still some confusing statements: Line 146, you do not anodize steel; Line 149, the surface is not steel surface; Line 210, what is anode film ? Line 218, no hard anodized film on steel but on aluminum; Line 239, not naked but uncoated; Line 253, you do not anodize steel but you coat it with Al that is anodized; Line 297, not densities but densifies, Line 298, not a color of steel but an Al-coated steel; Line 302, one last time, you have not anodized steel.
Reply: Thanks for reviewer’s detailed corrosion, we have stated Al-coated steel anodization instead of anodization steel surface.
(6) State that ASD means ampere per square decimeter.
Reply: Thanks for reviewer’s reminding, we have stated ampere per square decimeter for ASD means in line 182.
(7) Fig. 6 is misleading, you should plot current or current density and potential as a function of time.
Reply: We have plotted current as a function of time in Fig. 9. We have also added Fig.7 of V-T plot and descripted it in lines 197-203.
(8) Eq 3 and 4, is efficiency related to charge, a plot would be interesting.
Reply: Thanks for reviewer’s suggestion, we have added a plot of anodization efficiency related to charge, as Figure 8 and discussed it in lines 214-221.
(9) Fig 7 should have identical axes for all plots. What was the reason to use tests with different lengths? Did you aim for about 100 um calculated thickness?
Reply: In the present study, we use a final voltage of 75V as the baseline. This voltage is typical in practical application. Thus, “the anodization time and the film thickness were different for various ASDs” (revised in line 228).
(10) Lines 238-240 include some odd statements. Do you mean that the sprayed and rolled aluminium layer is about the same before anodization, but anodization itself cause corrosion on the steel/Al interface? In this case the rolled Al must be quite porous. If steel is anodically dissolved you should detect iron on the surface of Al, try the ferroxyl test.
Reply: we have revised the corresponding paragraph. We have added “It should be pointed out that there may still have defects in the thermally sprayed and hot-rolled Al film. If the applied current density is too high, it may induce poor anodization film with low hardness.” in lines 258-260.
(11) The caption of Fig 10 is misleading, especially figures c to e.
Reply: Thanks reviewer’s reminding, we have corrected it in lines 287- 288. Corresponding sentences concerning the figure is also revised (lines 272-273).
(12) Lines 274-275, how can you make electrochemical measurements in gas atmosphere, you need your sample and other electrodes immersed in solution?
Reply: We put our test sample and HCl solution in a closed container, that we can get HCl gas to corrosive on the testing sample surface. Corresponding sentences were added in lines 290-294.
(13) Fig 12 a analysis is done wrong, in Tafle analysis you should extend the linear ranges to corrosion potential and read the intersect. The corrosion current density is not the minimum value of the plot. Fig 12b the y-axis is not showing log(Z) but Z. Fig 12 d the axes must have identical tick lengths so that the plot is not distorted. Fig 12 e is mislabelled, the outer time constant is the Al and the inner one steel.
Reply: Thanks reviewer’s reminding, we have corrected it in Fig. 14. And, in lines 304, 307-309, 314, and 318.
(14) EIS analysis: If your component values are in ohms to 100 ohms range, how can you get several kilo-ohms as in Fig. 12b ? This is possible only if your Rct goes to kilo-ohms. Usually, what you have marked as Rct is actually the resistance in the pores of a coating.
Reply: In Fig 14(b), the impedance values are between 34 and 138 ohm in AISI steel bode plot, however, the impedance value are K-ohms in anodization bode plot. From fig. 14(c) Rct, rust is 7 ohm and Rrust is 135 ohm. From fig. 14(d) Rct, rust is 7 ohm and Rrust is 12000 ohm.
(15) Your conclusions about corrosion resistance are rather weak, you can expect aluminum to stand better corrosion than steel in HCl environment. The important part that is missing is the effect of different anodization processes on corrosion resistance.
Reply: Thanks for reviewer’s suggestion. Actually all the Al-coated steel after anodization exhibited excellent anti-corrosion behavior. We have added sentences concerning this issue in lines 290-294 and 330-335. At results and discussion, lines 290-294 “It should be pointed out that the AISI 1045 steel with thermally sprayed Al and anodization with various ASD exhibited excellent anticorrosion performance. No corrosion phenomenon can be observed after testing by typical 3.5 wt.% NaCl solution. In order to reveal the anti-corrosive ability, the test samples and HCl solution were placed in a closed container for various duration.” In conclusion, lines 329-334.
“Compared to Al-coated carbon steel surface with hard anodic film, the naked steel surface began to rust and color changed to brown after 30 min in HCl gas. Whereas all the Al-coated steel with anodization films can indeed improve the corrosion resistance of steel surface. No corrosion phenomenon can be observed after 24h in HCl gas environment.”

Round 3
Reviewer 2 Report
The 3rd version of the paper is acceptable.